# Improving Physical Activity Levels and Psychological Variables on University Students in the Contemplation Stage

**DOI:** 10.3390/ijerph16224368

**Published:** 2019-11-08

**Authors:** Cristina Corella, Javier Zaragoza, José Antonio Julián, Víctor Hugo Rodríguez-Ontiveros, Carlos Tomás Medrano, Inmaculada Plaza, Alberto Abarca-Sos

**Affiliations:** 1Faculty of Social and Human Sciences, University of Zaragoza, 44003 Teruel, Spain; criscor@unizar.es (C.C.); jajulian@unizar.es (J.A.J.); 2Centre for the Promotion of Physical Activity and Health (CAPAS-City), University of Zaragoza, 22001 Huesca, Spain; zaragoza@unizar.es; 3Faculty of Human Sciences and Education, University of Zaragoza, 22003 Huesca, Spain; 4Polytechnic School of Teruel, University of Zaragoza, 44003 Teruel, Spain; victor_rod_on@thotmail.com (V.H.R.-O.); ctmedra@unizar.es (C.T.M.); inmap@unizar.es (I.P.)

**Keywords:** university students, physical activity, intervention, self-determination theory, transtheoretical model

## Abstract

This study aimed to investigate the effects of a physical activity intervention, based on self-determination theory and the transtheoretical model, on university students in the contemplation stage. Participants: 42 students, in the contemplation stage at baseline, were randomly assigned to an experimental group (16 women, 2 men; M age = 19.1 ± 1.15) and a control group (18 women, 2 men; M age = 20.1 ± 5.7). Methods: Physical activity was measured at different moments by accelerometry. Other cognitive variables were measured by self-reported scales. Results: We did not find any significant increases in students’ physical activity in favor of the intervention group. Intragroup analyses indicate that the intervention has an effect on physical activity (moderate-to-vigorous physical activity), basic psychological needs, and intrinsic and extrinsic motivation. Conclusions: Results partially demonstrate that applying social cognitive theories seems to be effective in improving physical activity and cognitive variables in university students in the contemplation stage

## 1. Introduction

Current World Health Organization (WHO) guidelines recommend that adults aged 18–64 years should perform at least 150 min of moderate-intensity aerobic physical activity (PA), at least 75 min of vigorous-intensity aerobic PA, or an equivalent combination of moderate-to-vigorous intensity physical activity (MVPA) per week, to obtain multiple health benefits [1,2].

Despite the known benefits of PA [3], recent research findings have shown that childhood and adult PA levels have decreased over the past ten years in developed countries [4]. Trend data from the Spanish Health Survey [5] show that adherence to PA recommendations is fairly low among young people.

More specifically, there has been a drop in PA practice in young people aged between 18 and 24 years old [6,7,8], which is much more acute in the university student age range [9]. University students also form a population sub-group that is at risk of being sedentary, since a significant proportion of their time is spent studying or in class [10]. For example, Leighton and Swerissen [11] have shown that young adults attending university are far less active at university compared to the previous year, or months leading up to their arrival at university. Other studies indicate a continuing decline in PA levels as students’ progress from first year through to their later university years [12]. Specifically, 60% to 72.6% of Spanish university students did not meet PA guidelines [9,13].

Changes in lifestyle typical of the transit from school to university [13] may explain these figures. This can generate the appearance of new habits, greater pressure on academic results [14], and therefore a need to reconstruct their leisure time [15]. Further, changing to the university education system generates a lack of motivation for PA [16], and an increase in sedentary time [17]. Other factors that may influence this decrease in PA are the non-existence of physical education classes that could help to comply with MVPA practice recommendations [18], or the lack of adaptation of the offer of proposed activities to this population group [19].

The university environment has been considered an appropriate setting for health promotion initiatives [20,21]. Moreover, intervention programs have addressed different behaviors, giving special importance to PA practice [22], because it has demonstrated a prevention effect of several chronic diseases, even volumes of PA are minor than recommendations [3].

However, previous PA interventions have shown modest effects [23,24,25,26]. A growing body of evidence suggests that interventions, developed with an explicit theoretical structure that aim to change people’s behavior, are more effective than those lacking such a theoretical foundation [27]. In light of the complex and dynamic nature of PA behavior, it seems unlikely that a single theoretical approach can truly capture an individual’s motivations and behavioral patterns. The integration of theoretical frameworks allows researchers to achieve parsimony between constructs and identify the most powerful predictors of health behavior. Self-determination theory (SDT) and the transtheoretical model (TTM) provide promising support to promote initiation of various health behaviors, including PA [28,29,30].

The TTM [31] provides a framework for both the conceptualization and measurement of behavior change, and also facilitates promotion strategies. The stage of change (SOC) construct characterizes the time or readiness dimension into five progressive stages through which behavior change occurs: precontemplation (sedentary, no intention), contemplation (sedentary and 6-month intention), preparation (irregularly active and intention), action (regularly active for the last 6 months), and maintenance (regularly active for longer than 6 months) [31]. Although TTM has been recognized as a powerful model for behavior change, some limitations regarding the application of TTM to promote PA have been noted. For example, PA is a complex behavior and therefore, individuals may perceive the TTM constructs differently. Further, there are a number of PA influences (e.g., perceived competence) that are not accounted for by TTM [32].

SDT is a comprehensive theory of behavioral motivation, which indicates that both autonomous and controlled motivation may have an influence on behavior [33]. This theory may be a good reference framework to design intervention strategies geared towards this population sector [33], as the variables associated with motivation have great predictive power when explaining PA intention and behavior in the university population [34]. SDT is the social-cognitive theory that explains the greatest amount of variance in PA [35], and has also been proved to be effective in increasing PA levels in a wide range of populations [36].

Indeed, the SDT model may complement the TTM since it captures how intentional behavior originates and focuses on the internal change process, while the TTM captures the external change process [37]. The TTM framework holds that individuals in higher stages-of-change are more motivated than those at lower stages. SDT, on the other hand, places emphasis on the quality of motivation [38].

Thus, the aim of the present study was to examine the effect of a PA intervention, based on SDT and TTM strategies, among university students in the contemplation stage. In this regard, it was hypothesized, that, (hypothesis 1 (H1)) this intervention would significantly improve the PA level in the inter-group analysis, and (hypothesis 2 (H2)) this intervention is expected to improve satisfaction of the autonomy, competence and relatedness needs, intrinsic motivation, and also PA levels, in the experimental group.

## 2. Materials and Methods

### 2.1. Participants and Study Design

A quasi-experimental design was carried out. Figure 1 shows the flow of participants and the study design.

A total of 772 university students (168 males and 604 females; M age = 19.74 ± 42.76 years old) from the University of Zaragoza’s Teruel (Spain) Campus, were invited to participate in the study. Subjects completed a questionnaire at three different moments to identify their current status in relation to PA behavior (pre-contemplation, contemplation, preparation, action, or maintenance). 123 students, in the contemplation stage, were initially selected and finally, a total of 42 students, 37 women and 5 men (M age = 19.6 ± 4.2 years old) provided informed consent to participate in this intervention study. Participants were randomly assigned to the experimental group (EG) (19 women, 2 men; M age = 19.1 ± 1.15) and to the control group (CG) (18 women, 3 men; M age = 20.1 ± 5.7).

### 2.2. Intervention Program

The development of this intervention was based on guidance from an integrated framework of TTM and SDT, and evidence from effective health-related interventions in this population [39]. The intervention took place over 20 weeks (between October 2015 and May 2016), and it was divided into two different phases, cognitive (7 weeks, with a total of eight 60 min sessions) and behavioral (13 weeks, with a total of thirty 60 min sessions). During the intervention period, participants in the control group were instructed to continue with their normal PA habits.

The main objectives of the cognitive phase were: (a) awareness-raising, self-evaluation and environmental reevaluation, in relation to PA; (b) to examine the relationship between benefits (pros) and costs (cons) of PA practice; (c) to improve self-efficiency, in other words, the participants’ beliefs regarding their ability to successfully tackle the practice of physical education.

Meanwhile, the objectives of the behavioral phase were: (a) to facilitate knowledge and engagement in different PA practices that participants expressed their interest in; (b) to favor participants’ autonomous management of PA practice; and (c) to teach how to manage the APPtiva application, especially designed for this project, for them to be able to use it to daily monitor their physical activity, and to verify their degree of compliance with recommendations. Moreover, and to favor PA practice at weekends, different physical activity proposals were sent to each participant by email every week. A detailed description of all intervention strategies is available in Table 1 and Table 2.

Three strategies were used to encourage adherence to the intervention: (a) PA monitoring, using the App designed for the intervention; (b) encouraging participants to engage in PA practice at weekends. To this end, different PA proposals were sent to them by e-mail every week; and (c) three focus groups were conducted (i.e., before the intervention and after each of the phases), with the number of participants in each group ranging from 6 to 8 individuals. All focus groups followed a semi-structured discussion guide. The main focuses of the discussion were the participants’ goals, feelings and priorities, intervention barriers and facilitators, to explore their experiences and some key components of the intervention.

Measurement periods (i.e., objective PA outcome), were completed at three time points (e.g., before the intervention, after cognitive and behavioral phase). In addition, EG participants completed a questionnaire to measure Basic Psychological Needs (BPN) and motivation to PA.

### 2.3. Instrument and Measures

**Physical activity**. Physical activity was directly measured using accelerometers (Actigraph GT3X y GT3X+) at 10 s epoch setting [40].

Participants were instructed to wear the devices during waking hours, for 7 full days. Inclusion criteria to analyze data were wearing them for 10 h or more on 4 out of 7 days [40]. Non-wear time within a day was classified as an interval of at least 10 min of zero activity intensity counts, allowing for 1–2 min. of counts between 0 and 100 [41]. Data reduction was conducted using cut-points validated in youth [42]; 0–99, ≤1951, 1952–5724, 5725–9498, and ≥9499 counts per min^−1^ for sedentary, light, moderate, vigorous, and very vigorous activity, respectively.

**Stages of change (SOC)**. These were assessed using an ordered categorical SOC scale [43], showing appropriate construct validity [44]. The scale contains five items regarding a person’s PA habits, and intention to raise PA level. Participants had to select one of the items related to the stages: pre-contemplation, contemplation, preparation, action, or maintenance. Because the items are categorical, the present study did not yield Cronbach’s alpha.

**Basic Psychological Needs**. We measured participants’ perception of psychological need satisfaction, using the Psychological Needs Satisfaction in Exercise Scale (PNSE) [45]. This scale was validated in Spanish adolescents [46]. PNSE displays a number of psychometric characteristics that render the instrument useful for examining psychological need satisfaction in PA contexts. Internal consistency for subscale scores was estimated in our study (Cronbach α = 0.80 for competence, Cronbach α = 0.79 for autonomy and Cronbach α = 0.90 for relatedness).

**Motivation**. Sport Motivation Scale (SMS) [47] was used to measure PA motivation. It is based on the principles of SDT. This scale has proven to have satisfactory psychometric properties [48]. Specifically, the Spanish version also showed satisfactory internal consistency [49]. Our study presented an internal consistency of α = 0.84 for the dimension of intrinsic motivation, α = 0.91 for extrinsic motivation and α = 0.75 for amotivation.

**Fulfillment of PA recommendations**. We developed a mobile application (APPtive), so that participants in the intervention group could log in their MVPA minutes and fulfillment of PA recommendations [50].

### 2.4. Procedure

The research project was approved by the Clinical Research Ethics Committee of Aragon (Spain).

The intervention was implemented by different members of the research team. Participants in the experimental group (EG) and the control group (CG) completed the questionnaire at the university, supervised by the research team. Questionnaires were entirely voluntary, and all students were previously informed about the purpose of the study. Anonymity and confidentiality were guaranteed, in order to lessen social desirability bias, through the use of matriculation numbers rather than individual names and addresses.

Prior to the intervention, the research team discussed practical ways of how to implement the different strategies and activities in order to promote participants’ feelings of competence, of being connected to the other group members, as well as feeling more autonomous in their participation.

Participants were provided with a WhatsApp or e-mail contact for use during the entire intervention implementation process, so they could contact the research team regarding any doubts or problems.

Certain criteria (i.e., design of intervention) were considered to ensure the evaluation of the fidelity of intervention implementation. In this study, design of intervention was ensured by describing the content of each intervention session in detail.

### 2.5. Data Analysis

First, we analyzed the distribution of the sample through the Kolmogorov–Smirnov test, with results of *p* > 0.05 for the studied variables, concluding that the distribution was non-normal, so non-parametric statistics were used to calculate the differences.

Then, we examined the differences between the three times of the variables studied, both in EG and in CG, using the Wilcoxon test for one sample. Next, the Mann–Whitney U test was used to analyze if there were significant intergroup differences, and the Rosenthal r test [51] for non-parametric statistics was used to calculate effect size. Finally, the descriptive data of the variables, mean and standard deviation, were also provided. The “Statistical Package for the Social Sciences” software (SPSS IBM Inc., Chicago, IL, USA) version 21, was used in all analyses, considering *p* < 0.05.

## 3. Results

The pre- and post-intervention results showed, as seen in Table 3, that there were no significant differences in MVPA, when comparing the EG with the CG, between Time 1 (September 2015, just before starting the cognitive intervention), Time 2 (December 2015, coinciding with the end of the cognitive intervention phase), and Time 3 (May 2016, after the end of the behavioral phase).

Intragroup analyses have not shown significant differences in MVPA in either group.

There was a significant increase in the EG between the three times and in the different SDT variables (Table 4), more specifically, in basic psychological needs, with the exception of relatedness (between Times 2 and 3), and also in intrinsic motivation (in the three times) and in extrinsic motivation (between Times 1 and 2, and between Times 1 and 3).

## 4. Discussion

The purpose of this study was to evaluate the effect of an intervention based on constructs from the SDT and TTM. It was hypothesized that (H1) this intervention would significantly improve the PA level in the inter-group analysis (EG vs. CG), and (H2) this intervention is expected to improve satisfaction of the autonomy, competence, and relatedness needs, intrinsic motivation, and also PA levels, in the experimental group. Although the EG reported more minutes of MVPA, and improved students’ motivational outcomes after the intervention, the inter-group analysis indicated no significant differences between EG and CG at PA level.

These results are not surprising. A few PA interventions have targeted college students, and the overall results are not very encouraging [52]. Previous PA studies have shown modest effects on the initiation of PA [33,53], and other studies have previously observed [23,54,55,56] that only 64% of the interventions in this population had significant results [56]. Different reasons could explain our results.

First, behavioral change involves diverse factors. TTM suggests that behavioral change occurs in time through mechanisms that include cognitive and behavioral processes, meaning that motivation is decisive to pass from one stage to another [57]. It is possible that behavioral changes, unlike changes in cognitive factors, require more time, above all in individuals who wish to go from the contemplation stage to the preparation or action stage. People in the contemplation stage may remain so for a long time [58]. Hence, in future studies, a follow-up should be incorporated that will permit analyzing the post-intervention PA practice behavior. Intervention should include evaluation of outcomes following the end of intervention (optimally 12 months or longer) [59]. This could explain why the intervention has only been effective at intragroup level both in MVPA and in all cognitive variables (i.e., autonomy, competence and motivation), except for relatedness, possibly due to the fact that the intervention was implemented when the subjects had already established relatedness among each other. Cognitive variables have been identified as “active ingredients” of PA changes [60]. This is supported by SDT [27], which indicates that an improvement in BPNs related to PA, could significantly contribute to autonomous motivation in PA [61,62]. Further, the inclusion of self-regulation techniques, in particular self-monitoring of behavior (i.e., use of the APPtive), could be associated with improved PA effectiveness in the EG [63,64].

Second, the duration of the intervention was relatively short, above all taking into account that behavior changes need time and include several stages. Interventions spanning a university semester or less (≤12 weeks) have generally resulted in a greater number of significant outcomes in comparison to interventions with a duration of more than a semester [56]. However, as recommended by Prochaska and colleagues [57], to achieve behavior change, interventions should last for at least 6 months. The duration of the cognitive phase may also have been short, especially considering that in the transition from the contemplation stage to the preparation stage, cognitive processes were found to peak in the contemplation stage, and behavioral processes have proven to steadily increase from precontemplation to action, at which point they level off [65]. Probably, as the results of the present study suggest, more time is needed for university students to perceive PA as habit of healthy living [66].

Third, the participants were overwhelmingly female. This raises doubts about the approaches and strategies used in the intervention. Indeed, studies suggest that interventions targeting health behaviors in university students may need to be gender-specific to address the different needs and interests of both sexes [67]. Suggestions are made for PA professionals to incorporate gender differences in behavioral change intervention programs [68]. Therefore, customized PA programs and strategies are required to promote PA in female college students [52].

Fourth, the attendance of the EG individuals to the different intervention sessions could condition their effectiveness [69]. Despite applying different strategies to encourage adherence to the intervention, only 55.5% of the EG individuals participated in more than 50% the intervention sessions. Perhaps, to improve this adherence, PA practice (i.e., PA both on weekdays and at weekends) with others should be increased, because university students preferred PA with others outside of a structured class [70]. For this reason, the benefits of sharing activities among colleagues should be seriously considered by intervention policies to promote PA in this specific population.

Five, the measurement moment [71], more specifically, the last measurement (i.e., May 2016) coincided with exam time, and this situation could undoubtedly condition dedication to PA practice, as, during this period, the university students’ lifestyle could change [72]. The academic pressure that university students are subject to during this time [14] could re-orientate their time towards study, abandoning all other types of activities, such as PA practice.

Finally, the existence, within the contemplation stage, of different sub-stages [73] (i.e., “middle contemplation”, “pre-preparation”, “early contemplators”). Hence, the intervention may have had an influence on the fact that certain individuals have gone from one sub-stage to another within the actual contemplation stage without the existence of any significant increase in PA. According to Prochaska et al. [74], individuals may have become aware of the benefits of change, but they have still not committed to that change. Improvements in variables such as perceived competence over time could favor the change from one sub-stage to another, within the contemplation stage [73]. For example, the difference between individuals situated in the “middle contemplation” and “pre-preparation” sub-stages, is that the latter have greater competence and low reported cons of PA. Research indicates that competence for PA increases as individuals progress through the stages of change, and those competence scores correlate highly with stages of change [75]. Awareness of the existence of sub–stages within the contemplation stage may help to design and implement more precise intervention strategies. We suggest personalizing the intervention. For example, “early contemplators” may benefit most from interventions targeted at raising the pros of exercise and competence. Those in “pre-preparation” may benefit most from the provision of actual opportunities to practice PA [76].

### Limitations

Some limitations should be noted. First, the characteristics and limited sampling used. The small group size was due to the difficulties of conducting an intervention study in a structured educational context. Our results cannot be generalized to other areas in Spain, or abroad, as our participants came from one particular social and environmental academic context in Spain. Further, the participants included in this study may not be representative of all individuals in the contemplation stage. However, the results give an idea of the approach to be used to promote PA in this group. Second, participants’ perceived autonomy support, BPNs, and PA stages, were determined by surveys. Self-reported behavior measures are vulnerable to cognitive, affective, and self-presentational biases, which can lead to inaccuracies in behavior data [77]. However, self-reported PA questionnaires remain the method of choice for this type of assessment, since they are inexpensive, require little time, and are less likely to influence behavior [78]. Compared to previously published studies, one of the major strengths of this study was the inclusion of objective measures of the PA behavior variable. Objective measures should be the measurement of choice as they will provide more accuracy for the measurement of PA [78]. Another strength was the integration of different theoretical frameworks, and cognitive and affective strategies, to support the design and intervention planning in university students, in the transition from the contemplation to the preparation stage, compared with previous studies. In addition, another important strength of the study was the inclusion of self-monitoring PA behavior (e.g., APPtive). Applications, which are used on mobile phones, could be a better method to improve people’s lifestyles [64].

The findings of the present study have practical applications for PA specialists and other professionals who are involved in the development of large-scale PA programs at universities. The focus of such programs should be to incorporate both cognitive and behavioral strategies into an intervention process that must last for at least six months, if we really want to improve PA levels, and also incorporate post-intervention assessment processes. Further, a follow-up should be incorporated that will permit analyzing post-intervention PA practice behavior.

## 5. Conclusions

This intervention, based on strategies from TTM and SDT, was not effective in improving PA levels as shown by the comparison between the experimental and control groups. Further studies focusing on different combined cognitive and behavioral strategies, and with longer intervention duration are needed to identify possible differences between the experimental and control groups. Our study found a promising result in the improvement of MVPA and cognitive variables (i.e., autonomy, competence, and motivation) in the intervention group.

## Figures and Tables

**Figure 1 ijerph-16-04368-f001:**
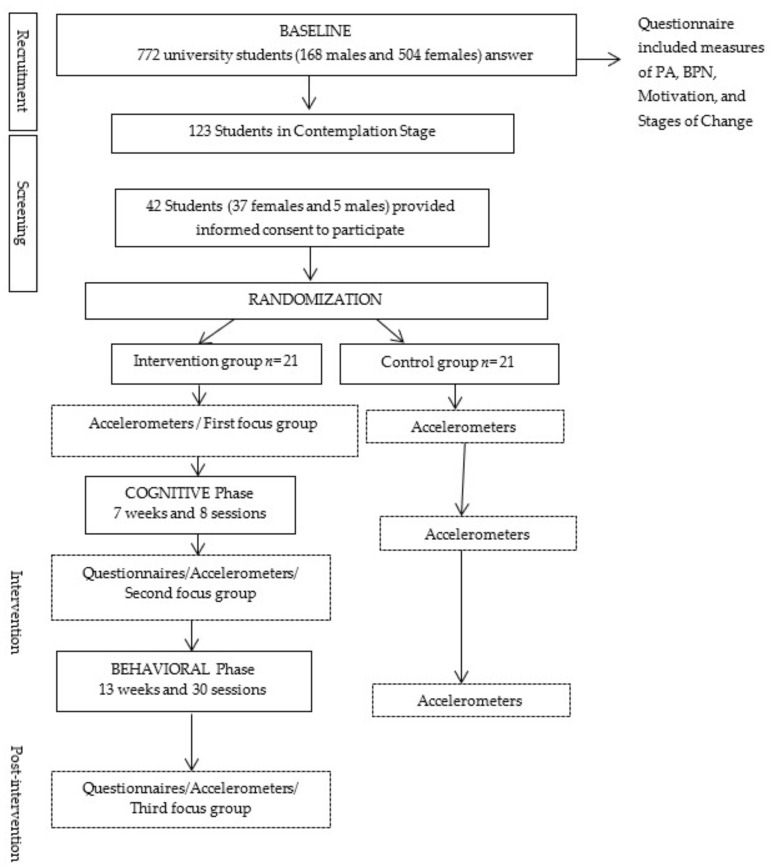
Flow of participants and study design. Abbreviations: Physical activity (PA); Basic Psychological Needs (BPN).

**Table 1 ijerph-16-04368-t001:** Intervention program strategies. Cognitive phase. Self-determination theory (SDT), transtheoretical model (TTM).

Strategies	Description	TTM/SDT	Primary Target
Awareness-raising	Meetings to increase information about the benefits and risks of physical activity	TTM	Awareness
Evaluate decisional balance	Motivational interviews to analyze pros and cons of changing behavior, identifying barriers; to identify intrinsic goals; to agree upon an action plan	TTM	Reevaluation
Time management	Use of time resources and the way to achieve goals. Regulation and control of time	SDT	Autonomy
Increase knowledge about physical activity guidelines	Participants receive information about the PA recommendations	SDT	Motivation
Knowledge about facilities	Provide knowledge about the facilities and activity offer of the city and the university	SDT	Autonomy
Emphasize responsibility	Researchers encourage participants to assume responsibility in decision-making	SDT	Autonomy

We acknowledge that the three needs are interrelated, and thus, strategies may support more than one need. However, we have highlighted the primary need to be targeted. Abbreviations: Physical activity (PA).

**Table 2 ijerph-16-04368-t002:** Intervention program strategies. Behavioral phase.

Strategies	Description	TTM/SDT	Primary Target
Choice of activities	Participants are given the option to choose different physical activities	SDT	Autonomy
Teamwork environment	Instructor focuses on teamwork	SDT	Relatedness
Progression in physical activity practice	Progression from simple to more complex skills and activities	SDT	Competence
Task climate	Provide a task-oriented climate during physical activity practice. Use positive reinforcement to recognize effort and progress	SDT	Competence
Group cooperation	Group participation in a popular race organized by the city council	SDT	Relatedness
Self-monitoring of behavior	Self-monitoring of outcome, providing informational feedback (use APPtiva)	SDT	Motivation
Support and encouragement	Send messages (WhatsApp) to provide feedback and advice	SDT	Motivation
Encourage physical activity practice alternatives during the weekend	Create programs and challenges for the weekend	SDT	Motivation

We acknowledge that the three needs are interrelated, and thus, strategies may support more than one need. However, we have highlighted the primary need to be targeted. Abbreviations: Trantheoretical Model (TTM); Self-determination Theory (SDT).

**Table 3 ijerph-16-04368-t003:** Intra- and inter-differences in MVPA in Times 1, 2, and 3.

	Time 1	Time 2	Time 3	Z (*p*) T1-T2	Z (*p*) T1-T3	Z (*p*) T2-T3
Experimental group ^a^	60.50 (20.34)	48.28 (21.95)	56.78 (23.61)	−1.992 (*p* = 0.813)	−0.628 (*p* = 0.181)	0.447 (p = 0.316)
Control group ^a^	50.74 (18.51)	43.09 (11.93)	42.51 (19.13)	−0.943 (*p* = 0.262)	−0.785 (*p* = 0.210)	−0.241 (*p* = 0.057)
Differences between experimental and control ^b^	Z = −1.178 (*p* = 0.208)	Z = −0.255 (*p* = 0.051)	Z = −1.906 (*p* = 0.344)			

^a^ Wilcoxon test to verify if there were differences between the MVPA means, between the three times of the experimental group, on the one hand, and of the control group, on the other. ^b^ Mann–Whitney U test to verify if there were differences between the means of the experimental and the control groups for each one of the three times. Note: The r corresponds to the effect size of the differences, calculated through Rosenthal r (1991). No significant differences were found for any value. Abbreviations: moderate-to-vigorous intensity physical activity (MVPA).

**Table 4 ijerph-16-04368-t004:** Experimental Group differences in BPN and motivation variables.

Experimental Group ^a^	Time 1	Time 2	Time 3
Perceived competence	3.63 (1.00)	4.39 (0.92)	4.72 (0.98)
Z (p): T1-T2; T2-T3; T1-T3 ^b^	3.72 (*p* < 0.001); 2.06 (*p* < 0.05); 3.72 (*p* < 0.01)		
r: T1-T2; T2-T3; T1-T3 ^c^	0.878	0.488	0.878
Autonomy	4.06 (1.12)	4.54 (0.89)	4.93 (0.86)
Z (p): T1-T2; T2-T3; T1-T3 ^b^	2.32 (*p* < 0.01); 2.18 (*p* < 0.05); 3.37 (*p* < 0.01)		
r: T1-T2; T2-T3; T1-T3 ^c^	0.547	0.515	0.796
Relatedness	3.48 (1.12)	4.25 (0.73)	4.41 (0.94)
Z (p): T1-T2; T2-T3; T1-T3 ^b^	2.98 (*p* < 0.01); 0.93 (*p* > 0.05); 3.68 (*p* < 0.001)		
r: T1-T2; T2-T3; T1-T3 ^c^	0.704	0.221	0.868
Intrinsic motivation	3.85 (1.34)	4.71 (1.02)	5.19 (1.01)
Z (p): T1-T2; T2-T3; T1-T3 ^b^	3.54 (*p* < 0.01); 2.22 (*p* < 0.05); 3.24 (*p* < 0.001)		
r: T1-T2; T2-T3; T1-T3 ^c^	0.791	0.524	0.764
Extrinsic motivation	2.84 (1.03)	3.34 (0.99)	3.65 (1.19)
Z (p): T1-T2; T2-T3; T1-T3 ^b^	2.21 (*p* < 0.05); 1.74 (*p* > 0.05); 3.19 (*p* < 0.01)		
r: T1-T2; T2-T3; T1-T3 ^c^	0.519	0.406	0.753
Amotivation	2.46 (1.04)	2.51 (0.81)	2.52 (0.96)
Z (p): T1-T2; T2-T3; T1-T3 ^b^	1.42 (*p* > 0.05); 1.32 (*p* > 0.05);0.52 (*p* > 0.05)		
r: T1-T2; T2-T3; T1-T3 ^c^	0.033	0.031	0.127

^a^ Wilcoxon test to verify if there were differences between the means of the different variables measured with questionnaires between the three measurement times for the experimental group. ^b^ the data for the Wilcoxon Z and significance are provided in the following order: differences between Time 1 and Time 2, differences between Time 2 and Time 3, and differences between Time 1 and Time 3. ^c^ The r corresponds to the effect size of the differences, calculated through the Rosenthal r (1991). Abbreviations: Basic Psychological Needs (BPN).

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
