# Peer review of "Improving Physical Activity Levels and Psychological Variables on University Students in the Contemplation Stage"

_ijerph, 2019, doi:10.3390/ijerph16224368_

Round 1

Reviewer 1 Report

please have a look at the uploaded file

Author Response

Kind regards

Reviewer 2 Report

Overall, it seems that the Authors rely on the general feeling coming from not so numerous studies, that interventions based on theoretical models are more effective. Not discussing is or is not true, their justification of the applied models and the construction of intervention targeting physical activity originates from rather old references and requires a significant update and more references adequately supporting their statement. The most representatives problems with providing appropriate bulk of evidence are discussed below. Apart from reaching for TTM and SDM models, we would expect a small review of modern approaches to stimulation of physical activities in various population groups.

In the Introduction, the statement about “a growing body of evidence …” (page 2) should be supported by relevant number of references (adequate to the statement of “a growing body”) apart from the position No 25.

The statement about “more promising theories of behavior change…” (page 2) should be also supported with adequate references showing if and why the theories are really promising. Now, the references No 26 and 27 are only about the model themselves without clear evidence why they are so promising.

Also if Authors believe that specifically TTM and SDM models are really promising in relation to physical activity, adequate set of references should be provided too. Authors use the references dated 2004 (reference No 35) and 2005 (No 36) to support their ascertainments that combined TTM and SDM models will be more effective in obtaining benefits in terms of physical activity interventions. They are quite ancient as for modern standards. Nothing happened in this area from 2005? If not, why to dig this artifacts? Apart from these reservations, both references are focused more on the constructs used in both models than on actual effect of intensity of physical activity.  So we still lack the argument for integration of these models in promoting physical activity.

In the Methods, the Authors mention that the used the strategy to encourage physical activity  practice during weekend in EG. It seems that CG was devoid of such opportunity. This is potentially the source of bias in assessing the results and understanding the usefulness of applied models.

In the Results section, first paragraph on page 7: “Intragroup analysis showed the existence of significant differences between times 2 and 3 in the EG with respect to MVPA”. It is not clear what the basis for such statement is. In the table 3, the comparison of T2 and T3 in EG does not show a significant difference (if we read the comment below the table 3).  Apart from explaining this contradiction, the p values should be transparently reported in the manuscript.

The authors should consider the correction for multiply comparisons in their inter- and intragroup calculations.

Taking into consideration the results presented in the table 3, the statement in Discussion and Conclusions that the results “partially” demonstrate effectiveness of intervention based on SDT and TTM on physical activity as measured by MVPA are rather puzzling. This must be corrected.

As it is provided now in the manuscript, the only types of outcomes demonstrating significant changes are those related to the applied model itself. This provides the basis for the statement that the model is effective in relation to outcomes expected from the model but not about physical activity which was inferred in the title of the paper.

The reviewer is not a native speaker, but experience with the type of design as in the manuscript prompts the suggestion that linguistic service would be necessary to obtain appropriate quality of the paper. 

Author Response

Kind regards

Round 2

Reviewer 1 Report

The title I find way too long; and it does not contribute to the selling of the manuscript.

Author Response

Dear reviewer,

The title I find way too long; and it does not contribute to the selling of the manuscript.   We have exchanged the tittle:  

Improving physical activity levels and psychological variables on university students in the contemplation stage

Kind regards

Reviewer 2 Report

Thank you for added text and modifications. Although Authors have visibly made an effort to adhere to Reviewer’s comment, I would still suggest some changes in tone and phrasing of some statements.

First of all, in the text added on the page 2, starting from „In light of the complex…” to „”… in University students], there a repetition of the adjective „promising”. I’m afraid using of such descriptive sound like the expression of the overoptimism of Authors in relation to discussed models and theories. Still, the evidence provided by Authors on feasibility and evidence of this theories is rather Modest. I would propose that this paragraph should be phrased in more realistic tone.

The statement starting from „Previous systematic reviews and meta-analysis…” on the same page is actually supported with only ONE systematic review (SR). So, please either provide one more SR or modify this statement that there is at least one systematic review with meta-analysis which …. etc.

Thank you for adjusting Discussion in relations to effect or their absence in relations to the outcomes used in the studies. The discussion of potential reasons for the discrepancy between changes in the model outcomes and no significant change in physical activity level is appreciated.

I would still suggest adding a paragraph to the Introduction including a mini-review of competing models and theories and strategies used to increase physical activity level in targeted audiences. As the references provided by the Authors in relation to .. are not to many, a reader may  feel somewhat surprised as there are generally many initiatives ongoing focused on increased physical activity. Maybe, some systematic review-type papers are available analyzing what type of rationale is used for implementing health programmes focused on physical activity?

The discrepancy between the Results and the Discussion sections in interpreting the outcomes of the analysis was clarified. Furthermore, the Authors corrected relevant table.

Author Response

Kind regards
